# First Experimental Evidence for Reversibility of Ammonia Loss from Asparagine

**DOI:** 10.3390/ijms23158371

**Published:** 2022-07-28

**Authors:** Jijing Wang, Sergey Rodin, Amir Ata Saei, Xuepei Zhang, Roman A. Zubarev

**Affiliations:** 1Department of Medical Biochemistry and Biophysics, Karolinska Institutet, SE-171 65 Stockholm, Sweden; jijing.wang@ki.se (J.W.); sergey.rodin@surgsci.uu.se (S.R.); amirata.saei.dibavar@ki.se (A.A.S.); xuepei.zhang@ki.se (X.Z.); 2Department of Surgical Sciences, Uppsala University, SE-751 05 Uppsala, Sweden; 3Department of Cell Biology, Harvard Medical School, Boston, MA 02115, USA; 4Endocrinology Research Centre, 115478 Moscow, Russia; 5Department of Pharmacological & Technological Chemistry, I.M. Sechenov First Moscow State Medical University, 119146 Moscow, Russia

**Keywords:** deamidation, isoaspartate, anti-aging, reverse reaction, mass spectrometry

## Abstract

Ammonia loss from _L_-asparaginyls is a nonenzymatic reaction spontaneously occurring in all proteins and eventually resulting in damaging isoaspartate residues that hamper protein function and induce proteinopathy related to aging. Here, we discuss theoretical considerations supporting the possibility of a full repair reaction and present the first experimental evidence of its existence. If confirmed, the true repair of _L_-asparaginyl deamidation could open new avenues for preventing aging and neurodegenerative diseases.

## 1. Introduction

Deamidation is a spontaneous, nonenzymatic loss of ammonia (NH_3_) from asparaginyls (_L_-Asn) and, to a lesser extent, glutamyl residues in proteins. Asn deamidation as well as water loss from aspartyls (_L_-Asp) in proteins result in succinimidyl (Succ). The latter cyclic group, being an unstable intermediate, quickly attaches a water molecule to become a β-isomer of Asp (isoaspartyl, isoAsp) [1,2]. IsoAsp differs from Asp by the rearrangement to the backbone of a CH_2_ side chain group, which distorts the secondary protein structure and can hamper protein function [3]. Moreover, deamidated proteins, such as human serum albumin (HSA), tend to lose their native conformation and start to aggregate [4,5], eventually contributing to the onset of neurodegenerative diseases [6,7,8].

Nature has designed several ways of dealing with deamidation (Figure 1a). In most organisms, the enzyme protein-_L_-isoaspartyl (_D_-aspartyl) *O*-methyltransferase (PIMT) can methylate isoAsp residues, using *S*-adenosylmethionine (SAM) as a cofactor [9,10,11]. Upon spontaneous methanol loss, methylated isoAsp converts back to Succ, which, upon water attachment, can convert to _L_-Asp in a minority (<25%) of cases. In the literature, such an inefficient two-stage process is usually referred to as isoAsp repair [10,12]. However, this process is more of a quick fix, as the final product of this “repair” is not the original _L_-Asn residue, but the biochemically different _L_-Asp. Here, we will call this process “partial repair” to distinguish it from the conceivable true repair, i.e., the conversion of isoAsp back to _L_-Asn.

Asn deamidation leading to isoAsp formation is ubiquitous, and it is one of the most common post-translational modifications (PTMs) in proteins. For example, in shotgun proteomics, Asn deamidation always occurs during sample preparation, typically during the trypsin digestion step that is performed overnight at 37 °C. Therefore, any competent proteomics data processing includes Asn deamidation as a variable modification [13]. However, likely because of the mundane chemical origin of isoAsp, there is a low awareness among biochemists of the existence and biochemical significance of deamidation. A notable exception is practicing biotechnologists, for whom Asn deamidation in recombinant proteins is a major nemesis to fight in their daily work [14,15].

Does the “quick fix” of Asn deamidation in the form of the PIMT + SAM “repair” of isoAsp represent a semi-efficient wrench in nature’s toolbox, or it is a first step of true repair leading to _L_-Asn? Until now, the mechanism of the latter process has remained unknown, and even its existence has not been postulated in literature. Furthermore, there is no enzyme known to catalyze Asn deamidation in humans (unlike glutamyl deamidation [16]), with only such deamidases found in viruses [17,18]. If there were such an enzyme in human cells, its reverse reaction would restore Asn from Succ or isoAsp by ammonia ligation.

## 2. Results and Discussion

Is the true repair of Asn deamidation possible, and if yes, does such a process occur in nature? From the theoretical perspective, one big obstacle in converting isoAsp or Succ to _L_-Asn could conceivably be the large enthalpy change associated with such a reaction. However, high-level simulations showed that even though the conversion of Asn to Succ and Succ to isoAsp involves overcoming large energy barriers, the overall enthalpy change in each of these steps is close to zero [19].

Of course, there are entropy considerations speaking against full repair. Since the concentration of ammonia (NH_3_) in the cellular environment is orders of magnitude lower than the water concentration [20], spontaneous NH_3_ attachment to Succ to form asparaginyl is far less likely than water attachment to form (iso)aspartyl. However, the microscopic reversibility principle still dictates that both a direct reaction (NH_3_ loss + H_2_O attachment) and a reverse reaction (H_2_O loss + NH_3_ attachment) take place. Therefore, at some level of deamidation, equilibrium specific to a given aspartyl and experimental conditions would be reached. Indeed, we found that in the artificial deamidation of HSA, Asn deamidation in the partial sequence LVNTEFAK saturates at ≈60%, with this level not changing between 28 and 42 days of incubation under deamidation-promoting conditions [21]. We also analyzed the deamidation degrees in other HSA asparaginyls and found that all of them reached the maximum after four weeks of incubation (Figure 1b). Therefore, it stands to reason that both the deamidation of Asn and the reverse activity (ligating NH_3_ to Succ or directly exchanging -OH to -NH_2_ from isoAsp) are possible via as yet unknown enzymatic activity. We named such a hypothetical enzyme protein succinimide/isoaspartate ammonia ligase (PSIAL). Isoaspartate is added to the enzyme name because the one-step conversion of isoAsp to Asn is also possible, as a one-step Asn deamidation process leading to isoAsp has been postulated [22].

If PSIAL exists, it should be active in cells that are known for their self-renewal properties, such as stem cells, including cancer stem cells. We embarked on a search of the evidence for PSIAL activity, and here report the first promising results.

First, we designed an assay for detecting the PSIAL activity, consisting of a mixture of artificially deamidated peptide LVGNGVL containing 99% isoAsp (Figure 2a,b), with added SAM to assist partial isoAsp repair, ATP as a source of possibly needed energy, and a nitrogen-rich amino acid. This mixture was added to a tested cell lysate and incubated for 4 h at 37 °C and, in parallel, at 4 °C for control. After incubation, the mixture was filtered through a 3 kDa membrane and the filtrate was analyzed by LC-MS/MS (Figure 2c). The peptide LVGNGVL in three forms (with Asn, Asp, and isoAsp, Figure 2b) was quantified as customary in label-free proteomics, and the fold-change α of the relative abundance of the Asn form in 37 °C incubation versus 4 °C was taken as a measure of the PSIAL activity. All experiments were performed in *n* ≥ 3 replicates.

In the first round of experiments on the HCT116 cell lysate, the most significant PSIAL activity was obtained using 100 µM SAM and 100 µM _L_-Gln as nitrogen-rich amino acid (α = 7.8 ± 0.8; Figure 2d and Appendix A). Importantly, using 100 µM _D_-Gln instead of _L_-Gln did not show a PSIAL effect (α = 0.9 ± 0.1), as would be expected if _L_-Gln were a nitrogen source for isoAsp repair. The relative PSIAL activity at 100 µM SAM and 100 µM _L_-Gln was measured for lysates of A498, RKO, HCT116, A549, MCF-7, and HT29 cell lines (Figure 2e), with the highest activity found for HT29 cells. To localize the PSIAL activity, we separated the lysed HT29 cells into cytoplasmic, membrane, nuclear, chromatin-bound, and cytoskeletal fractions. The highest PSIAL activity was detected in the cytoplasmic fraction (α = 1.9 ± 0.2), and the lowest in the nuclear fraction (α = 1.2 ± 0.1, Figure 2f).

To verify these findings, we performed a second series of experiments with artificially aged HSA (aHSA) as a deamidated polypeptide (Figure 3A–C), incubating it with SAM, ATP, and nitrogen donors (Figure 3D). The ETD MS/MS spectrum confirms the isoAsp presence by the z’_8_-57 ion (Figure 3C). As a source of nitrogen, Gln was added in either isotopically normal form or as ^15^N-enriched in either the amide or amine groups. The tryptic peptide LVNTEFAK and its deamidated variants produced in trypsin digestion of aHSA were monitored by LC-MS/MS. Besides the higher abundance of the Asn form, its molecular mass increase was expected if isotopically normal ammonia was lost from the asparaginyl of LVNTEFAK, and then ^15^N atoms in ammonia (from both Amine-^15^N and Amide-^15^N Gln) was added to Succ or -^15^NH_2_ was exchanged with –OH in isoAsp. The relative mass shift was monitored as a centroid of the mass distribution for an isotopic cluster of the MH^+^ molecular ion encompassing its monoisotopic peak M as well as two subsequent isotopic peaks M + 1 and M + 2. The results provided statistically significant evidence for PSIAL activity (*p* < 0.05), with the largest mass shift occurring with amide ^15^N (Figure 3E). Both Amine-^15^N and Amide-^15^N Gln contributed to the reamination reaction, which could be due to the existence of multiple reamination enzymes. In addition, since deamidation is a slow process, the repair may also be slow, and slow-acting enzymes may be less specific as they explore a wide range of different molecular configurations over a long time, each with its own specificity. Moreover, the specificity of enzymes is often overestimated, while for evolutionary reasons they must preserve wide specificity [23].

As the highest PSIAL activity was obtained in largely soluble cytoplasmic proteome fraction (Figure 3C, we performed thermal proteome profiling experiments [22] on cell lysate to determine which proteins interact with an artificially deamidated peptide Gly-isoAsp-Gly-isoAsp-Gly-isoAsp-Gly-isoAsp-Gly, using the _L_-Asp version of the same peptide as a control. Among the 2796 proteins with well-fitting melting curves (*p* < 0.05) from three independent experiments, cytoplasmic aspartate aminotransferase (GOT1) was at the top of the list based on the combination of the melting temperature shift ∆T and *p* value of the shift (Table 1). To verify PSIAL activity in that protein, recombinant enzymes GOT1 and, as a negative control, bleomycin hydrolase BLMH, were incubated with aHSA, ATP, SAM, and _L_-Gln at 37 °C and 4 °C, respectively. The cofactor pyridoxal 5’-phosphate known for GOT1 enzyme was also added to the GOT1 samples. The analysis supported PSIAL activity for GOT1, but not for bleomycin hydrolase, as expected (Figure 3F). The α for GOT1 was 1.26, which is a relatively small value, probably due to the slow kinetics of cellular deamidation and the reversed repair reaction.

Summarizing, here we present the first experimental evidence that Asn deamidation in proteins may be reversed through enzymatic activity. The combined evidence appears to surpass the threshold for discovery. Among the six cancer cell lines, the highest PSIAL activity was found in HT29 cells known to contain a high percentage of cancer stem cells [24]. If confirmed by further research, the true repair of _L_-asparaginyl deamidation could open new avenues to preventing deamidation-related aging and diseases (e.g., neurodegenerative diseases), thus prolonging healthy lives.

## 3. Materials and Methods

### 3.1. Artificial Aging of the Synthetic Peptide

The synthetic peptide LVGNGVL (SynPeptide) was incubated in 50 mM Tris buffer (pH 8.5) at 50 °C for 18 days. The aged peptide was then analyzed by an Easy-nLCII high-pressure liquid chromatography (LC) system (Thermo Fisher Scientific, Waltham, MA, USA) coupled with a Q Exactive HF mass spectrometer (Thermo Fisher Scientific, Waltham, MA, USA). The Easy-nLCII system was equipped with a self-packed column (100 μm × 200 mm, 3 μm C8 beads (Dr. Maisch, Ammerbuch, Germany)). Peptide elution was performed using a binary solvent system consisting of 0.1% (*v*/*v*) formic acid, 2% (*v*/*v*) acetonitrile (solvent A) and 98% acetonitrile (*v*/*v*), 0.1% (*v*/*v*) formic acid (solvent B) with a flow rate of 300 nL/min. The gradient was of 10–35% B in 20 min followed by 35–95% B in 3 min. The isoaspartate (isoAsp) occupancy was 99.9% after 18-day incubation, and this sample was used as “aged peptide” in the following PSIAL activity experiment.

### 3.2. PSIAL Activity Experiment on “Aged Peptide”

One million of the HCT 116 cells (ATCC, *p* = 6) were washed with cold phosphate-buffered saline (PBS) (Cytiva, Marlborough, MA, USA) three times. The cell pellet was dissolved in PBS buffer supplemented with EDTA-free protease inhibitor (Roche, Basel, Switzerland) and phosphatase inhibitor (Roche, Basel, Switzerland), snap-frozen in liquid nitrogen, and thawed at room temperature. After repeating the “freeze and thaw” procedure three times, the cell suspension was centrifuged at 14,000× *g* and the supernatant was collected as cell lysate. The mixture in six replicates was prepared with 20 μL of each component in PBS: 10 μM “aged peptide”, 1 mg/mL cell lysate, 100 μM ATP, together with nitrogen donor (_L_-Gln or _D_-Gln) and S-adenosylmethionine (SAM) with different concentrations (as shown in Figure 2a). Three replicates were incubated at 37 °C for 4 h, while the rest of the replicates were incubated at 4 °C for 4 h as controls. All samples were filtered using the Amicon Ultra 3 kDa filter (Millipore, Burlington, MA, USA) and the filtrate was analyzed via mass spectrometry as described above. The fold-change α of each sample was calculated as the relative abundance of the Asn form (P1) at 37 °C incubation versus 4 °C.

### 3.3. PSIAL Activity Experiment in Different Cell Lines

Cell lysate from different cancer cell lines (A498, RKO, HCT116, A549, MCF-7, HT29) in the same passage (*p* = 6) was extracted from six replicates using the same procedures described above, and 1 mg/mL of each cell lysate was incubated respectively with the mixture as above. The fold-change α of each cell lysate was calculated.

### 3.4. PSIAL Activity Experiment in Subcellular Fractions

The five subcellular fractions (cytoplasmic, membrane, nuclear, chromatin-bound nuclear, cytoskeletal) were extracted from HT29 cell lysate using a Subcellular Protein Fractionation Kit (Thermo Fisher Scientific, Waltham, MA, USA) in six replicates, according to manufacturer’s instructions. Then, 1 mg/mL of different fractions were incubated respectively with the mixture of 10 μM “aged peptide”, 100 μM _L_-Gln, 100 μM SAM, and 100 μM ATP similar to the procedures above, and the fold-change α of each fraction was calculated.

### 3.5. Thermal Proteome Profiling–Temperature Range (TPP-TR) Experiments

To determine which proteins specifically interact with a short isoAsp-containing peptide, the thermal proteome profiling–temperature range (TPP-TR) experiments were performed according to the reported method [25]. In short, the specific melting temperatures for each protein from the “treatment” and “control” samples were compared. Here, the peptide GisoAspGisoAspG (SynPeptide, Shanghai, China) was used as a “drug” and GDGDG (SynPeptide, Shanghai, China) as a “control” to treat the HCT116 cells. The other procedures were followed as previously described [1]. The temperature shift of each protein was calculated, as well as the *p*-value between two groups. For proteins with a *p*-value less than 0.05, we generated the score using the formula: $Score=-\;|\mathrm{Median}△T|\times {\mathrm{Log}}_{10}\;(P\;\mathrm{Value}\;\mathrm{of}\;△T)$. 

### 3.6. Artificial Aging of Human Serum Albumin (HSA)

The HSA (Sigma Aldrich, Saint Louis, MO, USA) was incubated in 50 mM Tris buffer (pH 8.0) at 60 °C for 42 days, digested to peptides, and analyzed by an LC-MS/MS, as described [2]. The isoAsp occupancy of the peptide sequence LVNTEFAK showed the highest (~60%) among the asparagine-containing peptides. The “aged HSA”, after 42 days of incubation, was used in the following PSIAL activity experiment.

### 3.7. PSIAL Activity Experiment on “Aged HSA”

The 0.1 mg/mL “aged HSA” in eight replicates was mixed with 1 mg/mL HT29 cell lysate, 100 μM SAM, 100 μM ATP, and 100 μM nitrogen donor (Amide-^15^N-enriched Gln, Amine-15N-enriched Gln, or normal Gln), and each component was prepared in 20 μL PBS. After incubation of 37 °C and 4 °C for 4 h, respectively, the samples were subjected to a nano-LC system Ultimate 3000 (Thermo Fisher Scientific, Waltham, MA, USA) connected in-line with a Fusion Orbitrap mass spectrometer (Thermo Fisher Scientific, Waltham, MA, USA). Peptide separation was performed on a 50 cm long EASY spray column (PepMap, C18, 3 μm, 100 Å) with a gradient of 5–30% B in 75 min, and 30–95% B in 5 min (Gradient A: water with 2% acetonitrile and 0.1% formic acid, and Gradient B: acetonitrile with 2% water and 0.1% formic acid). Precursor ion fragmentation was performed with simultaneous higher-energy collision dissociation (HCD; collision energy: 27%, resolution 70,000, AGC target 5.0e4, maximum injection time 200 ms) and electron-transfer/higher-energy collision dissociation (EtHCD; collision energy: 40%, resolution 17,500, AGC target 5.0e4, maximum injection time 200 ms). The relative mass shift of each sample was calculated according to the formula:$$\mathrm{Relative}\;\mathrm{Mass}\;\mathrm{Shift}\;(\%)=\frac{Intensity\;of\;\left(M+1\right)+2\times Intensity\;of\;\left(M+2\right)}{Intensity\;of\;M}\times 100\%.$$

## Figures and Tables

**Figure 1 ijms-23-08371-f001:**
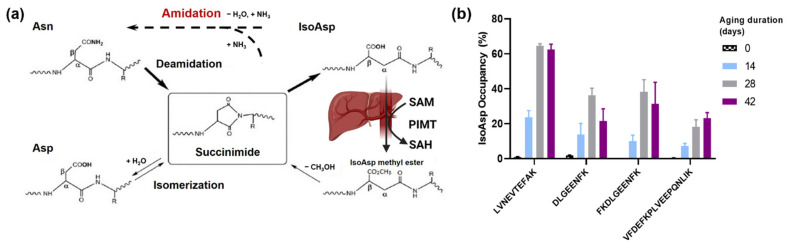
**Proposed protein succinimide/isoaspartate ammonia (PSIAL) activity.** (**a**) The mechanism of isoAsp formation and repair by PMIT and the proposed amidation via enzymes with PSIAL activity. (**b**) The deamidation degree of asparaginyl peptides in HSA after artificial deamidation for 42 days.

**Figure 2 ijms-23-08371-f002:**
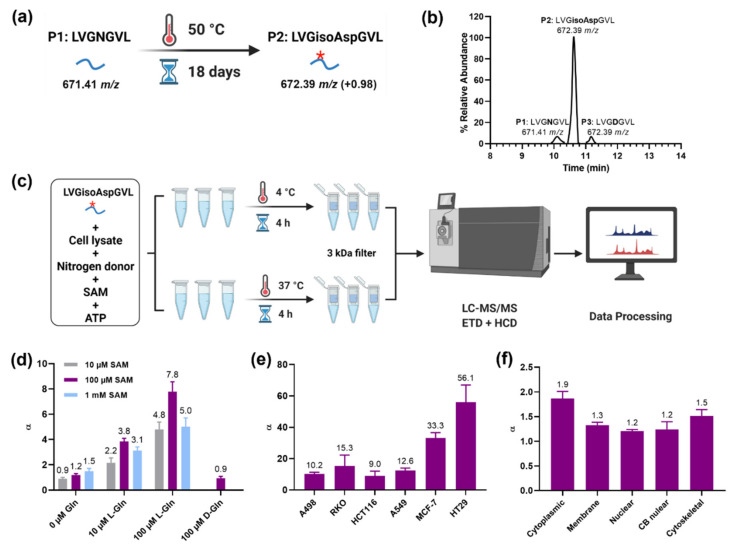
**Discovery of PSIAL activity on synthetic isoAsp-containing peptide.** (**a**) Artificial deamidation of the synthetic peptide LVGNGVL (P1) to LVGisoAspGVL (P2) via incubating at 50 °C for 18 days. *: isoAsp. (**b**) The chromatogram of the aged peptide above in three forms with Asn (P1), isoAsp (P2), and Asp (P3). (**c**) The scheme of testing PSIAL activity in cell lysate using aged P1 and cofactors. (**d**) The effects of cofactors (Gln and SAM) with different concentrations on PSIAL activity. (**e**) Comparison of PSIAL activities among different cancer cell lines by measuring the fold-change α of the relative abundance of the Asn form of peptide in 37 °C incubation versus 4 °C. (**f**) Comparisons of PSIAL activities among different subfractions of HT29 cell lysate.

**Figure 3 ijms-23-08371-f003:**
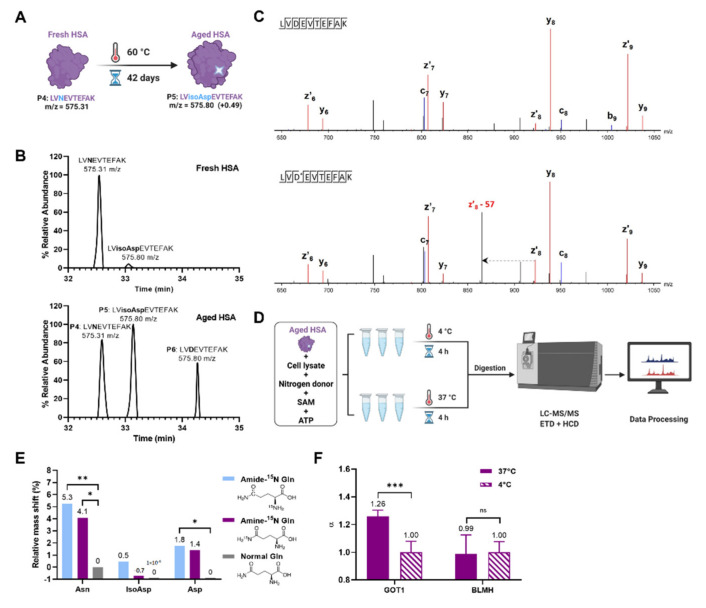
**Discovery of PSIAL activity on aged HSA.** (**A**) Artificial deamidation of fresh HSA to aged HSA via incubating at 60 °C for 42 days. (**B**) The chromatogram of fresh (**up**) and aged (**down**) HSA (D* = isoAsp). (**C**) ETD MS/MS spectrum confirms the isoAsp presence by the z_8′_-57 ion (**up**), and its absence in the corresponding peptide with Asp (down). (**D**) The scheme of testing PSIAL activity in cell lysate using aged HSA and cofactors. (**E**) The mass shift in the Asn form of an HSA reporter peptide after adding Amine-^15^N _L_-Gln, Amide-^15^N _L_-Gln, and isotopically normal _L_-Gln. (**F**) Validation of PSIAL activity in GOT1 protein with BLMH as a negative control. * *p* < 0.05, ** *p* < 0.01, *** *p* < 0.001.

**Table 1 ijms-23-08371-t001:** Top 10 protein candidates interacting with isoAsp in a peptide.

Protein ID	Protein Name	Gene Name	|Median △T|	−Log_10_ (*p* Value)	Score *
P17174	Aspartate aminotransferase, cytoplasmic	GOT1	9.8	4.6	45.6
P36959	GMP reductase 1	GMPR	9.6	3.1	29.9
Q9UFN0	Protein NipSnap homolog 3A	NIPSNAP3A	12.4	2.1	25.6
E2QRF9	Geminin	GMNN	7.7	2.6	20.3
O94992	Protein HEXIM1	HEXIM1	7.7	2.4	19.0
Q13126	S-methyl-5-thioadenosine phosphorylase	MTAP	4.3	4.2	17.9
P41236	Protein phosphatase inhibitor 2	PPP1R2	10.2	1.7	17.4
P61956-2	Small ubiquitin-related modifier 2	SUMO2	8.1	2.0	16.3
O43852	Calumenin	CALU	7.6	2.1	16.2
P00390-2	Glutathione reductase, mitochondrial	GSR	8.6	1.8	15.5

* $Score=-\;|\mathrm{Median}△T|\times {\mathrm{Log}}_{10}\;(P\;\mathrm{Value}\;\mathrm{of}\;△T)$

## Data Availability

Not applicable.

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
