# Peer review of "First Experimental Evidence for Reversibility of Ammonia Loss from Asparagine"

_ijms, 2022, doi:10.3390/ijms23158371_

Round 1

Reviewer 1 Report

The article by Wang et al addresses a very innovative question in the field of protein deamidation. This reaction is described for almost a century as the one-way loss of the amino group from the lateral chain of Asn and Gln residues in proteins. Only a partial repair is operated by the enzyme PIMT in cells, but the original L-residues are not restored. This article explores the possibility that a reaction might exist to operate the full repair.

From the general point of view, the article is well written, but some points need to be better explained, and others should even be corrected as will be outlined below in my comments. I have a few points of concern: one is that the authors discard some of the degradation byproducts from the general scheme of the reaction, and although I understand the rationale, a more compelling justification is needed; another one is that the authors are looking for an enzyme that will decrease the activation barrier of the reamination reaction, so that the enzyme-based catalysis will be more efficient than a chemical-based catalysis. The demonstration for the latter point is not convincingly established in my opinion.

Point by point analysis:

Some aspects of the reaction are overlooked:

- line 35: only the production of L-residues is considered while L-succinimide may also racemize into D-succinimide, the isomerization of which will produce D-IsoAsp (see works from Meinwald in 1986 and from Haley in 1966). What do the authors hypothesize regarding this point? Wouldn’t a “true repair” mechanism fix all the byproducts of the reaction (L-IsoAsp and D-IsoAsp)? or could D-IsoAsp be repaired by a different mechanism?

- Figure 1 describes Asn deamidation (and Asp isomerization) as proceeding only through the cyclic intermediate called succinimide. And the text lines 46-51 explicitly envisage a repair mechanism exclusively reversing deamidation through the cyclic intermediate. But deamidation under mildly acidic conditions lead to the direct hydrolysis of the amino group without the formation of succinimide. The groundings used by the authors to exclude a direct repair mechanism that does not go through succinimide should be better documented/explained.

Line 37: “Asn deamidation leading to isoAsp formation is ubiquitous, and the most common post-translational modification (PTM) in proteins.” This sentence may be confusing and could use some clarification + bibliographic reference. It cannot be disputed ex vivo, but in vivo, other PTM are certainly more common.

Some statements are misleading: line 49 “there is no enzyme known catalyzing Asn deamidation (unlike glutamyl deamidation)”. This is not true. The Fen lab identified viral deamidases both in gamma herpes viruses and alpha herpes viruses. Gamma HV deploys vGAT in infected cells, which assembles with a eukaryotic homolog to deamidate 2 Asn residues and one Gln residue of the protein RIG-I. Alpha HV deploy UL37, a functional deamidase which deamidates in vitro and in cells two Asn residues of RIG-I.

Line 50: “If there was such an enzyme, its reverse reaction would restore Asn from Succ by ammonia ligation”. This assumes that the deamidase operate through the same chemical steps as non-enzymatic deamidation (with the same reaction intermediates): this is certainly a possibility, but can we exclude that other reaction substates are generated, and if yes, on what grounding?

Figure 1b states that the deamidation degree of Asn peptides in HSA is saturated after artificial deamidation for 42 days. Even though two time points hardly define a plateau, I could agree for the first peptide, but the second and third peptide seem to peek at 28 days and decrease at 42 days, while the fourth seems to be constantly ascending. In order to maintain the strong statement written lines 73-74, another time-point should be recorded.

Lines 81 and 99: a correlation is suggested between effective repair of deamidation and cell stemness. This is purely speculative and a better use of this interesting idea would be in the perspective opened by this work rather than in the course of the demonstration where no experiments effectively relate to stemness.

Figure 2:

The experimental design of the article uses comparative data acquired at 4° and 37° from a mixture containing cell lysates as a source for the putative PSIAL enzyme. In order to better establish the argumentation between a chemical-based reamination and an enzyme-based reamination, the proper control should include heat-inactivated cell lysates rather than or in addition to operating at 4°C.

Regarding my earlier comment on the standpoint adopted by the authors that reamination proceeds through succinimide, wouldn’t it be appropriate in figure 2 to omit SAM from the assay and see whether the reaction proceeds differently?

From personal expertise, peptides and protein treated chemically to accelerate deamidation tend to undergo proteolysis: how is the total amount of the peptide verified at each endpoint in this experimental setup? Couldn’t proteolysis be the source of artifacts in the quantification?

Line 115 indicates that PIMT was added in the reaction mix, but this is neither mentioned in the caption, nor justified in the text. Please clarify.

Figure 3:

I may not have grasped the subtlety of Fig 3e, but what I understand is that Asn reamination can be fed both from Amine-15N Gln and from Amide-15N Gln. If my understanding is correct, then this strongly argues against an enzyme-based catalysis as enzymes are stereospecific. Please explain.

Also, I do not understand why Asp show a relative mass shift in this assay. Please explain.

Finally, Figure 3f shows that the relative “gain” of enzyme-based reamination relative to chemical-based reamination is 1.26. Could the author elaborate on this value and its efficiency toward the slow kinetics of cellular deamidation?

Table 1: I am curious of what is the take of the authors on the reason why so many (apparently) unrelated proteins are found to interact with the IsoAsp peptide?

Minor comments:

Figure 1: Caption reads “Proposed PISAL activity” instead of PSIAL activity. Please define the acronym in the text before referring to figure 1 because the explanation is only found line 76.

Some of the choices of words are a bit odd:

line 28: nature “prepared”

line 42: the “mundane” chemical origin of IsoAsp? please explain.

Author Response

Thanks for the comments and suggestions. Please see the response as attached file.

Point by point analysis:

The article by Wang et al addresses a very innovative question in the field of protein deamidation. This reaction is described for almost a century as the one-way loss of the amino group from the lateral chain of Asn and Gln residues in proteins. Only a partial repair is operated by the enzyme PIMT in cells, but the original L-residues are not restored. This article explores the possibility that a reaction might exist to operate the full repair.

From the general point of view, the article is well written, but some points need to be better explained, and others should even be corrected as will be outlined below in my comments. I have a few points of concern: one is that the authors discard some of the degradation byproducts from the general scheme of the reaction, and although I understand the rationale, a more compelling justification is needed; another one is that the authors are looking for an enzyme that will decrease the activation barrier of the reamination reaction, so that the enzyme-based catalysis will be more efficient than a chemical-based catalysis. The demonstration for the latter point is not convincingly established in my opinion.

Some aspects of the reaction are overlooked:

Point 1: - line 35: only the production of L-residues is considered while L-succinimide may also racemize into D-succinimide, the isomerization of which will produce D-IsoAsp (see works from Meinwald in 1986 and from Haley in 1966). What do the authors hypothesize regarding this point? Wouldn’t a “true repair” mechanism fix all the byproducts of the reaction (L-IsoAsp and D-IsoAsp)? or could D-IsoAsp be repaired by a different mechanism?

Response 1: It is an interesting question, but we have no definite answer. If D-isoAsp was a significant product, we would see four peptide peaks in LC-MS, but in all studied cases we saw only three, which means that racemization was not a big factor.

Point 2: - Figure 1 describes Asn deamidation (and Asp isomerization) as proceeding only through the cyclic intermediate called succinimide. And the text lines 46-51 explicitly envisage a repair mechanism exclusively reversing deamidation through the cyclic intermediate. But deamidation under mildly acidic conditions lead to the direct hydrolysis of the amino group without the formation of succinimide. The groundings used by the authors to exclude a direct repair mechanism that does not go through succinimide should be better documented/explained.

Response 2: We apologize for the misunderstanding. In Figure 1, we not only propose the repair mechanism of isoAsp via NH3 ligation to Succ, but also via H2O loss from + NH3 attachment to isoAsp (with two dotted arrows). We didn’t exclude the direct repair from isoAsp, and we add “or isoAsp” after “restore Asn from Succ” in Line 52. PSIAL means Protein Succinimide/Isoaspartate Ammonia Ligase, which means that enzymes with PSIAL activity convert either isoAsp or Succ back to Asn (Line 61, 78-80).

Point 3: Line 37: “Asn deamidation leading to isoAsp formation is ubiquitous, and the most common post-translational modification (PTM) in proteins.” This sentence may be confusing and could use some clarification + bibliographic reference. It cannot be disputed ex vivo, but in vivo, other PTM are certainly more common.

Response 3: Thanks. We have rephrased the sentence to “Asn deamidation leading to isoAsp formation is ubiquitous, and it is one of the most common post-translational modifications (PTMs) in proteins”.

Point 4: Some statements are misleading: line 49 “there is no enzyme known catalyzing Asn deamidation (unlike glutamyl deamidation)”. This is not true. The Fen lab identified viral deamidases both in gamma herpes viruses and alpha herpes viruses. Gamma HV deploys vGAT in infected cells, which assembles with a eukaryotic homolog to deamidate 2 Asn residues and one Gln residue of the protein RIG-I. Alpha HV deploy UL37, a functional deamidase which deamidates in vitro and in cells two Asn residues of RIG-I.

Response 4: Thank you for this important correction. We have revised the sentence and added the reference in Line 50-51.

Point 5: Line 50: “If there was such an enzyme, its reverse reaction would restore Asn from Succ by ammonia ligation”. This assumes that the deamidase operate through the same chemical steps as non-enzymatic deamidation (with the same reaction intermediates): this is certainly a possibility, but can we exclude that other reaction substates are generated, and if yes, on what grounding?

Response 5: Indeed, we cannot, but no such substates have been identified to the best of our knowledge. We have added “or isoAsp” after “ligating NH3 to Succ” in Line 76, and after “added to Succ” in Line 121.

Point 6: Figure 1b states that the deamidation degree of Asn peptides in HSA is saturated after artificial deamidation for 42 days. Even though two time points hardly define a plateau, I could agree for the first peptide, but the second and third peptide seem to peek at 28 days and decrease at 42 days, while the fourth seems to be constantly ascending. In order to maintain the strong statement written lines 73-74, another time-point should be recorded.

Response 6: This is an interesting suggestion, but for our purpose it is enough that the deamidation degree does not increase past certain time point. We rephrased the corresponding sentence.

Point 7: Lines 81 and 99: a correlation is suggested between effective repair of deamidation and cell stemness. This is purely speculative and a better use of this interesting idea would be in the perspective opened by this work rather than in the course of the demonstration where no experiments effectively relate to stemness.

Response 7: Thanks. We have revised the phrasing and moved the sentence to Line 151-152.

Point 8: Figure 2:

The experimental design of the article uses comparative data acquired at 4° and 37° from a mixture containing cell lysates as a source for the putative PSIAL enzyme. In order to better establish the argumentation between a chemical-based reamination and an enzyme-based reamination, the proper control should include heat-inactivated cell lysates rather than or in addition to operating at 4°C.

Response 8: We discussed this suggestion, but decided against the heat-inactivated cell lysates as control, since the heat would increase the risk to a) induce more deamidation; b) inactivate endogenous enzymes or cofactors needed for the reaction; c) affect the native structure and reduce the solubility of the substrates.

Point 9: Regarding my earlier comment on the standpoint adopted by the authors that reamination proceeds through succinimide, wouldn’t it be appropriate in figure 2 to omit SAM from the assay and see whether the reaction proceeds differently?

Response 9: We have the data for samples with 10 µM Gln without SAM, and they did not exhibit PSIAL activity – we now add this data to Supplementary information.

Point 10: From personal expertise, peptides and protein treated chemically to accelerate deamidation tend to undergo proteolysis: how is the total amount of the peptide verified at each endpoint in this experimental setup? Couldn’t proteolysis be the source of artifacts in the quantification?

Response 10: The artificial deamidation was done via incubation at 60 ËšC and pH 8.5; these conditions are not known to induce proteolysis. Also, proteolysis is inefficient at the isoaspartyl bonds and thus ongoing proteolysis would increase the relative abundance of the isoAsp peptide, while the opposite was observed.

Point 11: Line 115 indicates that PIMT was added in the reaction mix, but this is neither mentioned in the caption, nor justified in the text. Please clarify.

Response 11: We apologize for the mistake. We have revised the sentence in Line 116.

Point 12: Figure 3:

I may not have grasped the subtlety of Fig 3e, but what I understand is that Asn reamination can be fed both from Amine-15N Gln and from Amide-15N Gln. If my understanding is correct, then this strongly argues against an enzyme-based catalysis as enzymes are stereospecific. Please explain.

Response 12: Yes, we further explain it in Line 127-132. Both Amine-15N and Amide-15N Gln contributed to the reamination, which could be due to the existence of multiple reamination enzymes. Also, since deamidation is a slow process, repair may also be slow, and slowly acting enzymes may be less specific since over long time frame they explore a wide range of different molecular configurations, each with its own specificity. Moreover, the specificity of enzymes is often overestimated, while for evolutionary reasons they must preserve wide specificity, please see Peracchi A, The Limits of Enzyme Specificity and the Evolution of Metabolism, Trends in Biochemical Sciences 2018, 43(12) DOI:10.1016/j.tibs.2018.09.015.

Point 13: Also, I do not understand why Asp show a relative mass shift in this assay. Please explain.

Response 13: The mass shift may originate from reamination by 15NH3 Asn that subsequently deamidated by 14NH3 loss to form Asp.

Point 14: Finally, Figure 3f shows that the relative “gain” of enzyme-based reamination relative to chemical-based reamination is 1.26. Could the author elaborate on this value and its efficiency toward the slow kinetics of cellular deamidation?

Response 14: We think that as cellular deamidation is slow, reamination can also be slow. We have added the suggested phrasing in Line 145-146. As for chemical reamination in vivo, it is also a possibility, but the molecule described in the cited peptide seems to be toxic, and a less toxic biological analogue is not known.

Point 15: Table 1: I am curious of what is the take of the authors on the reason why so many (apparently) unrelated proteins are found to interact with the IsoAsp peptide?

Response 15: It’s pretty common in our experience to have many proteins that pass through the p-value filter. What is important is that GOT1 is an outlier even among these proteins, as its combined score is almost twice as high as that of the next-best candidate.

Point 16: Minor comments:

Figure 1: Caption reads “Proposed PISAL activity” instead of PSIAL activity. Please define the acronym in the text before referring to figure 1 because the explanation is only found line 76.

Response 16: Thanks for pointing out. We have revised accordingly.

Point 17: Some of the choices of words are a bit odd:

line 28: nature “prepared”

line 42: the “mundane” chemical origin of IsoAsp? please explain.

Response 17: Thanks for pointing out. We have revised the phrases above in the text.

Reviewer 2 Report

The authors present a straight forward manuscript that expands upon their hypothesis that there is likely an enzymatic reaction that protects the cell isoaspartate residues. The manuscript seems to follow the format of a short communication and that seems appropriate given the type of data presented. The pictures that accompany data figures are an excellent help for interpretation. 

Figure 1 legend: where the legend is placed the PSIAL term has not been identified and should be spelled out for clarity in its current format. the PSIAL term is typed incorrectly in Figure 1's legend.

Introduction in page 2, the 'Does the "quick fix" is a little unclear if the authors are providing a quote or if there is an addition character that was inadvertently included. 

Second sentence of Section 2. Results and Discussion needs to be examined closely as it reads a little unclear.

Author Response

Thanks for the comments and suggestions. Please see the response as attached file.

Point by point analysis:

The authors present a straight forward manuscript that expands upon their hypothesis that there is likely an enzymatic reaction that protects the cell isoaspartate residues. The manuscript seems to follow the format of a short communication and that seems appropriate given the type of data presented. The pictures that accompany data figures are an excellent help for interpretation. 

Point 1: Figure 1 legend: where the legend is placed the PSIAL term has not been identified and should be spelled out for clarity in its current format. the PSIAL term is typed incorrectly in Figure 1's legend.

Response 1: Thanks, we have added the abbreviation explanation for “PSIAL” in Figure 1 legend.

Point 2: Introduction in page 2, the 'Does the "quick fix" is a little unclear if the authors are providing a quote or if there is an addition character that was inadvertently included. 

Response 2: Thanks for pointing out. It is now corrected in Line 46.

Point 3: Second sentence of Section 2. Results and Discussion needs to be examined closely as it reads a little unclear.

Response 3: We have rephrased the description in Line 61.

Round 2

Reviewer 2 Report

Accept in current form.